# Assessing Different PCR Master Mixes for Ultrarapid DNA Amplification: Important Analytical Parameters

**DOI:** 10.3390/diagnostics14050477

**Published:** 2024-02-23

**Authors:** Ivan Brukner, Miltiadis Paliouras, Mark Trifiro, Marc Bohbot, Daniel Shamir, Andrew G. Kirk

**Affiliations:** 1Lady Davis Institute for Medical Research, Jewish General Hospital, Montreal, QC H3T 1E2, Canada; mark.trifiro@mcgill.ca; 2Department of Medicine, McGill University, Montreal, QC H3G 2M1, Canada; 3Nexless Healthcare, 1315 Chem. Canora, Mont-Royal, Montreal, QC H3P 2J5, Canada; marc@nexless.io (M.B.); dshamir@nexless.io (D.S.); 4Department of Electrical and Computer Engineering, McGill University, Montreal, QC H3A 0E9, Canada; andrew.kirk@mcgill.ca

**Keywords:** PCR, ultrafast, plasmonic PCR, molecular, diagnostics, DNA, technology, Taq polymerase

## Abstract

The basic principles of ultrafast plasmonic PCR have been promulgated in the scientific and technological literature for over a decade. Yet, its everyday diagnostic utility remains unvalidated in pre-clinical and clinical settings. Although the impressive speed of plasmonic PCR reaction is well-documented, implementing this process into a device form compatible with routine diagnostic tasks has been challenging. Here, we show that combining careful system engineering and process control with innovative and specific PCR biochemistry makes it possible to routinely achieve a sensitive and robust “10 min” PCR assay in a compact and lightweight system. The critical analytical parameters of PCR reactions are discussed in the current instrument setting.

## 1. Introduction

PCR-based diagnostics employ nucleic acid amplification to provide the greatest specificity and sensitivity for acute infection detection. However, conventional PCR machines are slow, taking hours to deliver results, and they are also bulky and power-consuming. These techniques have applications in fields such as clinical diagnostics, pathogen detection, and genotyping, where rapid turnaround times are crucial. Ultrarapid PCR techniques aim to provide results in a fraction of the time compared to traditional PCR methods [1,2,3,4,5,6,7,8].

The plasmonic PCR methodology uses a completely novel and simple way to generate and manage the heat needed for thermocycling, enabling it to be used for an ultrafast mode without losing sensitivity and specificity [9,10,11,12]. Plasmonic thermocycling is achieved using optically powered nano-heaters suspended in the PCR reaction mixture itself. Specifically, gold nanorods (GNRs), when exposed to laser light, can generate enormous heat in femtoseconds. Heating is instantaneous and extremely precise by cycling (on/off) a compact 808 nm laser. GNRs have exemplary light absorption properties (>90%) and are 100% efficient in converting absorbed light to heat, reaching rates of 20 °C/second. The plasmonic PCR device used in this study employs vertical cavity surface emitting lasers (VCSELSs) as laser sources and contains four independent PCR chambers. Temperature monitoring uses non-contact IR thermometry (up to 50 measurements/s). Since the tube is completely free, cooling is accomplished only through forced-air convection. Efficient and extremely rapid thermocycling allows for a highly specific PCR product (30 cycles in ~10 min, 20 μL volume). The device also enables incorporated real-time amplicon monitoring by using an ultra-sensitive fluorescent dye (SYTO-16), which produces Ct values and melting curves indistinguishable from conventional qPCR platforms. Finally, the has the potential to operate at any scale, and because the energy needs are vastly reduced, a self-powered portable version can be easily designed and demonstrated.

In this context, plasmonic PCR reaction, characterized by 1–2 s of denaturation and merged annealing–elongation steps (A/E), belongs to the category of ultrafast PCRs. To test these “ultrafast” plasmonic PCR reaction conditions, we compare two PCR mixes, Luna^®^ Universal qPCR & RT-qPCR (NEB, Boston, MA, USA) and Solis FAST^®^ SolisGreen^®^ qPCR Mix (Solis Biodyne, Tartu, Estonia), using the same PCR assay [13]. These PCR master mixes were exposed to extreme cycling conditions, where total amplification time was in the range of 10 to 30 min. General and specific comments about the limits of the analytical sensitivity, signal/noise ratio, reproducibility, and accuracy in detecting DNA targets with ultrafast PCR are documented. Additionally, the use of an optimized Taq-polymerase, Solis FAST^®^ SolisGreen, clearly demonstrates that the biochemistry of PCR reaction is an important factor in achieving the promised detection and speed limits of ultrafast plasmonic PCR.

## 2. Materials and Methods

### 2.1. Sample

For detecting the performance characteristics of different PCR master mixes, under extremely short cycling conditions, we used a 10× serial dilution of a COVID-19 DNA-positive control of the N gene (200,000 copies/µL) (IDT DNA, Coralville, IA, USA).

### 2.2. Primers

The qPCR assays were composed of published and previously evaluated primer combinations, with corresponding positive controls. The data in this report targeted the COVID-19 N gene, generated with forward and reverse primers 5′ACCCAATAATACTGCGTCTTGG3′ and 5′GGTAGCTCTTCGGTAGTAGCC3′ (250 nM) (IDT DNA) using temperature conditions as described by Sarkar et al. [13]. The holding time within each PCR cycle was 1 s denaturation (92 °C) and 15, 5, or 1 s (s) of A/E (merged annealing/elongation steps) at 60 °C.

### 2.3. PCR Master Mixes

The two enzyme master mixes we tested herein are recognized for their tolerance for rapid cycling conditions: (a) Luna Universal One-Step RT-qPCR Kit (NEB, Boston, MA, USA), named Luna assay further in the text, and (b) Solis FAST^®^ SolisGreen^®^ qPCR (Solis Biodyne, Tartu, Estonia), referred to as Solis assay. They are both capable of working with fluorescent intercalating dyes. The current technical requirements of our PCR instrumentation are such that we must use the SYTO-16 intercalating dye, characterized by a high quantum yield (Thermo Fisher Scientific, Saint-Laurent, QC, Canada).

Our PCR assay biochemistry has a common setting: the poly (ethylene glycol)-modified gold nanorods (PEG-GNRs) with a 50 nM particle concentration (purchased from Nanopartz, Loveland, CO, USA). The nanorods we used had an 811 nm longitudinal resonance wavelength and a 4:1 aspect ratio. An optimized 2.5 nM final concentration of PEG-GNP [9] was used for all assays developed.

### 2.4. Plasmonic PCR Instrument

The Kimera P-IV, VCSEL 4-channel plasmonic PCR instrument (Nexless Healthcare, Montreal, QC, Canada) was used to maximize performance and repeatability. All experiments presented herein used the same single channel out of the 4-channel instrument. The cycle threshold (Ct) and melting temperature profiles (Tm) were calculated using analytical software tools (Nexless Tm, V2.02) incorporated into the Kimera P-IV instrument. The melting profiles of the amplicons obtained in each post-PCR reaction tube were further validated using a CHAI Open QPCR System instrument (Santa Clara, CA, USA).

## 3. Results and Discussion

### 3.1. Thermal Cycling Profiles

During analysis of thermal cycling profiles, we did not observe changes in heating or cooling among tubes having different PCR master mixes. Therefore, we conclude that the PEG-GNR-SYTO-16 assay PCR conditions we used are the dominant factor in defining the thermal cycling profile, and not the master mix type. However, performance characteristics of the PCR reaction, measured by using DNA amplification descriptors such as the cycle threshold (Ct) commonly used in quantitative PCR (qPCR), or the DNA melting profile (Tm) of post PCR tubes, were significantly different among different PCR Master Mix Manufacturers, under the same instrumental setting.

Typically, the Taq-based DNA extension speed in PCR is ~60–100 base-pairs/s [14], usually measured through rapid deletion mutants of Taq (KlenTaq) [15], which are missing 5′-3′ exonuclease activity [2,16,17]. This hydrolyzing 5′-3′ Taq activity is a prerequisite for TaqMan probe-like assays, but the drawback of using these enzymes is the need for longer annealing/extension times. Herein, we compared two commercial PCR master mixes that do have 5′-3′ exo-activity; although compatible with probe-based assays, we assessed amplification using the SYTO-16 dye to measure the limits of detection between the two PCR mixes.

Performance features of Luna and Solis PCR assays were tested using 10-times serial dilutions on a Positive Control COVID-19 N gene plasmids, starting from approximately 10 gene-equivalents per tube. The tests covered a 4 log_10_ range (10 through 10,000 copies) of input DNA as assessed performance of the two enzymes over shortening AE times of 15 s, 5 s, or 1 s, through a measurement of Ct and post-PCR Tms (Table 1). For 15 s AE time, we found quite similar Ct and Tm results between the Luna and Solis assays. The difference among PCR master mixes became more obvious and repeatable between the shorter A/E times. For the Luna assay, this linearity of the Ct values deteriorated as the A/E time was further shortened (Figure 1). Due to the nature of the signal monitoring (fluorescence intercalation dye), primer-dimers might have contributed to the value of the Ct parameters together with the intended amplicon.

### 3.2. Melting Curve Analysis—Amplification Accuracy of Solis PCR Assay

Therefore, we proceeded with the analysis of the Tm values. From Table 1 distinct Tm values are detected, especially for lower template copy numbers. From our observations, the Tm values for the intended amplicon are 79.9 °C (Solis assay) and 78.1 °C (Luna assay). The negative controls (not shown) have noise-like, small picks around 71.9 °C and 63.3 °C (Luna and Solis assays, respectively). Although not obvious from the analysis of the Ct values, the Tm analysis is able to further differentiate the outperformance of Solis vs. Luna assays. In Figure 2, representative Tm curves of Solis and Luna, at 10,000 copies, show that the Luna assay begins losing performance at the 5 s A/E time, with the visualization of primer dimers. Solis is able to maintain ~80 °C Tm curve peaks at all A/E times. As such, at the extremely short 1 s A/E time, the Luna assay was consistently unsuccessful across all template concentrations, whereas the Solis assay still demonstrated successful PCR amplification.

As a measure of reproducibility (relative variability among replicated experiments), we used the Coefficient of Variation (CV) of numerical Tm values across triplicate experiments, with a lower CV indicating a higher reproducibility. The CV of the Tm values of the Solis assay for the intended amplicons were less than 1% across all A/E times and DNA template concentrations. The equivalent strategy for estimated PCR reproducibility has been documented [13,18,19]. Similarly, accuracy was considered as a capacity to discriminate Tm profiles among the intended amplicon and primer-dimers and/or negative controls. It is defined as the number of correct “predictions” versus total number of “predictions”. From Table 1, we can clearly see that the use of the Tm parameter guarantees maximal analytical accuracy, with the confident distinction between positive and negative PCR test results.

## 4. Conclusions

The above results indicate that the PCR assay can be accomplished in 10 min for 40 cycles with a high PCR sensitivity and impressive reproducibility among replicate experiments, within the same PCR reaction chamber. With the nested PCR approach used by Bio Fire-like protocols [20] and further design miniaturizations [2,13,20,21,22,23,24,25], this PCR could have impressive multiplexing capacity without any significant changes in biochemistry.

In summary, our analysis showed the following: (1) a post-PCR Tm analysis effectively distinguishes shorter amplicons (“primer-dimers”) from the intended ones. The Tm values for the intended amplicons were consistently a few degrees higher than for the non-intended products, a distinction that could be incorporated into clinical data analysis algorithms. (2) There is a critical difference in Taq polymerase performance among different PCR master mix vendors. (3) The plasmonic PCR reaction shows excellent reproducibility and accuracy in single-channel measurements. (4) The Solis assay’s engineered Taq polymerase retains 5′-3′ exonuclease activity, allowing for the use of TaqMan hydrolysis probes in PCR multiplexing. (5) Future real-world applications, using clinical samples for validation will be indispensable.

## Figures and Tables

**Figure 1 diagnostics-14-00477-f001:**
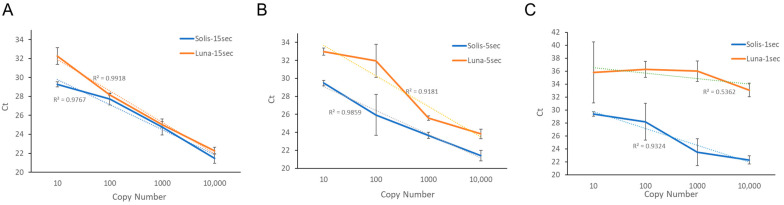
Graphical representation of Ct values of Luna vs. Solis. Ct values are averaged and plotted for each copy number concentration for 15 s (**A**), 5 s (**B**), and 1 s (**C**) A/E times. For each A/E time, the Solis PCR assay maintains a strong linear curve with an R^2^ above 90%.

**Figure 2 diagnostics-14-00477-f002:**
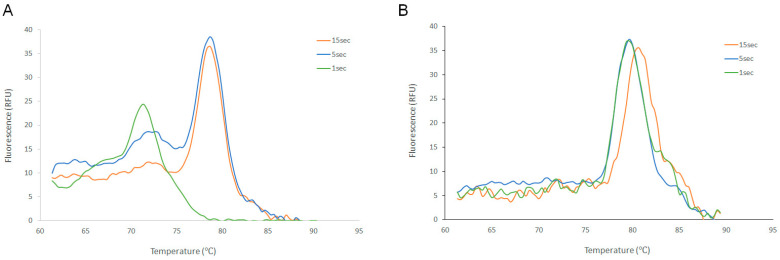
Melting profile of primer-dimers and intended target. Illustrated are Tms curves for Luna (**A**) and Solis (**B**) PCR assays amplifying 10,000 target copies at 15, 5, and 1 s A/E times. With Luna enzyme, at the 5 s A/E time, primer-dimers are seen at 72 °C along with the target amplification at 78 °C. However, for the Solis PCR assay, only the intended target amplicon Tm curves are visualized at all A/E times.

**Table 1 diagnostics-14-00477-t001:** The PCR-related Ct and Tm parameters of Luna vs. Solis assays.

Luna Universal qPCR & RT-qPCR												
Copy Number	**15 s A/E**				**5 s A/E**				**1 s A/E**			
	Ct-s	Av. Ct	Tm-s	Tm CV	Ct-s	Av. Ct	Tm-s	Tm CV	Ct-s	Av. Ct	Tm-s	Tm CV
10	31.35	32.26	71.7/78.5	0.29	32.9	32.97	71/76	n.d.	37	35.77	71.3	n.d.
	33.07		72.2/78.6		32.6		71.7		35.4		71.5	
	32.36		71.7/78.2		33.4		72		34.9		72.0	
100	28.22	28.17	77.0	0.33	31.47	31.96	73.5/78.5	0.18	37.3	36.27	71.3	n.d.
	28.05		76.8		33.7		70 *		36.6		71.1	
	28.24		77.3		30.7		72.5/78.3		34.9		72.5	
1000	24.7	25.04	77.9	0.46	25.35	25.59	77.83	0.51	37.53	36.04	73.8	n.d.
	25.15		78.5		25.82		77.8		36.19		72.5	
	25.28		78.5		25.75		78.5		34.4		73.3	
10,000	22.6	22.27	78.7	0.19	23.27	23.82	71.3/78	0.32	32.8	33.13	71.0	n.d.
	22.3		78.8		24.3		71.9/78.3		32.3		72.4	
	21.9		78.5		23.9		71/78.5		34.3		72.2	
0	n.d.	n.d.	72	n.d.	n.d.	n.d.	n.d.	n.d.	n.d.	n.d.	72.3	n.d.
	n.d.		71.6		n.d.		n.d.		n.d.		71.1	
	n.d.		72.1		n.d.		n.d.		n.d.		72.0	
**Solis FAST qPCR Mix**												
Copy Number	**15 s A/E**				**5 s A/E**				**1 s A/E**			
	Ct-s	Av. Ct	Tm-s	Tm CV	Ct-s	Av. Ct	Tm-s	Tm CV	Ct-s	Av. Ct	Tm-s	Tm CV
10	29.55	29.28	79.5	0.57	29.8	29.50	80	0.00	29	29.37	75/80	0.36
	29.3		79.9		29.4		74/80		29.7		74/80	
	29		79		29.3		74/80		29.4		74/79.5	
100	28.15	27.72	80	0.40	28.11	25.93	73/79	0.53	28.25	28.18	74/80	0.64
	27		80		23.57		79.33		31		74/81	
	28		79.45		26.11		72.8/79.8		25.3		80.3	
1000	24.18	24.78	79.1	0.94	23.41	23.66	80.8	0.67	21.96	23.52	80	0.32
	25.37		80.7		23.9		79.9		25.86		79.3	
	24.8		80		27 *		79.9		22.74		79.5	
10,000	21.3	21.45	80.3	0.66	21.08	21.39	79.9	0.57	21.61	22.25	79.7	0.48
	21.03		79.5		21.08		80.2		22.25		79.3	
	22.01		79.3		22		80.8		22.9		80.1	
0	n.d.	n.d.	63	n.d.	n.d.	n.d.	73	n.d.	n.d.	n.d.	73	n.d.
	n.d.		62.8		n.d.		63		n.d.		74	
	n.d.		63.2		n.d.		62.5		n.d.		62.5	

The Target numbers are approximations and presented as result of 10× serial dilutions, n.d. is “not determined”, 0 copy number is a no-template negative control reaction. For analysis of Tm, when two peaks are observed, both are noted. Furthermore, Tm Coefficient of Variance (CV) was not calculated for values that did result in a not-positive amplicon. * Value removed from analysis.

## Data Availability

Data is contained within the article.

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
