# Peer review of "Assessing Different PCR Master Mixes for Ultrarapid DNA Amplification: Important Analytical Parameters"

_diagnostics, 2024, doi:10.3390/diagnostics14050477_

Round 1

Reviewer 1 Report (Previous Reviewer 2)

Comments and Suggestions for Authors

The authors successfully replied to all comments.

Author Response

Review 1

Comments and Suggestions for Authors

The authors compared Luna Universal and Solis FAST®SolidGreen, two Taq-DNA ultrafast enzymes with 5'-3' exonuclease activity, for plasma® ultrafast PCR reactions, to regulate elongation time and N gene concentration parameters, and to evaluate PCR amplification results by Ct values and melting curves, redefining the speed and accuracy of PCR diagnosis. The results showed that the Ct and Tm results were similar at 15 seconds of annealing, and that the Luna assay was unsuccessful due to the extremely short annealing and elongation times of one second, while the Solis assay still demonstrated successful PCR. The use of Tm will guarantee maximum analytical accuracy at a value of 1 that reliably differentiates positive and negative PCR test results within the detection limit. The techniques used have been previously reported, but the correlation of Tm with PCR amplification results is rarely mentioned. Therefore, I support publishing it in diagnostics. Here are some of the key comments from this work:

  1. The authors must do a comprehensive review for the ultrafast plasmonic PCR in the Introduction, since the ultrafast PCR has been proposed and developed for many years. Some critical studies reported about the ultrafast PCR should be referred to, such as Light: Science & Applications2015, 4, e280; Trends in Biochemical Sciences2020, 45 (2), 174-175. –

We have provided a review of Plasmonic PCR as how it functions within the prototype device developed by the authors and Nexless Healthcare.   The basic principles of Plasmonic PCR are described in review Trends in Biochemical Sciences, the functional application of the performance of an actual plasmonic PCR device are first illustrated in this manuscript.    Furthermore, the research paper Light: Science & Applications 2015, 4, e280; uses a gold nanofilm with the PCR occurring on the surface of the film.  All the components of our PCR assay, including the gold nanorods, are incorporated into the reaction mixture.  Bulk temperature of the mixture is monitored, rather than the gold film surface, using uses non-contact IR thermometry.  Subsequently,  fluorescence is also monitored optically

  1. In this work, the photothermal material (gold nanorod), ultrafast enzyme and PCR instrument are all commercially available, and the experimental design is not innovative.

The PCR instrument described in this submission, at the time, was not commercially available.  Moreover, we worked with several vendors of enzymes and nanomaterial, to provide for optimal performance of the Plasmonic PCR instrument.  Through the development process, several things were discovered about the functioning of Plasmonic PCR that differentiates it from conventional platforms.  First and foremost, the ability to rapidly thermocycle required a change to the PCR protocol and selection of enzymes.  As we illustrate in our manuscript, the Luna PCR assay could not adequately amplify the low target concentrations when the A/E times were shortened to 5 seconds or 1 second.  Therefore, the application of Plasmonic PCR systems is not going to be a direct transition of PCR protocols from conventional platforms.

  1. The major concern is that the presented data is too little to support the results. The authors thus are suggested to add more data, including nanomaterials characterization, experimental condition optimization, ultrafast PCR performance etc.

We took considerable time and effort to provide better illustrations of our data.  The manuscript now includes two figures and a revised Table.  Figure 1, shows the change in and loss of linearity of Ct values as the A/E becomes shortened from 15sec to 1sec, between the two PCR assays.   Figure 2 provides representative Tm melting curves of Luna vs. Solis PCR assays, that illustrate the underperformance of Luna polymerase as the A/E times are shortened.  Moreover,  Figure 2 also supports our conclusions that the assessment of Tm is more informative than Ct values when making a final decision on the positive outcome of the reaction.

  1. The supplementary materials were linked to a company website and a Thesis of Master Degree, which is not proper for the published papers.

We have corrected this by removing the citation of the Master’ thesis, and have provided only published papers for our references.

Reviewer 2 Report (Previous Reviewer 1)

Comments and Suggestions for Authors

The authors have not answered my previous questions one-by-one.

Comments on the Quality of English Language

Minor editing of English language required

Author Response

Reviewer 2 

Real-time plasmonic PCR of the Covid-19 DNA N gene positive control DNA with 10-10,000 copies of input and 1-15 sec. of annealing/elongation (in triplicate) was performed using two different master mixes.

The following points should be considered:

  1. M&M, line 75-78. Why is this relevant for M&M? It may be relevant for the introduction, but not in this section.

Thank you, this has been corrected.

  1. M&M, line 92 and elsewhere. Reference 16 is a master thesis, which is hard to access for future readers.

Thank you, agree, The Masters’ thesis reference has been removed.

  1. Results and discussion, line 109-111. This is not described in M&M.

Sorry for this, has been corrected.

  1. Results and discussion, line 113-114. There are no results documenting this.

This section of the manuscript has been revised and placed in the Materials & Methods of the manuscript, which describes the PCR protocol and the concentration of the mixture components. 

  1. Results and discussion, line 117-120. Please refer to table 1.

Thank you, discussion has been referred to Table 1.

  1. Results and discussion, line 141-142. The linear change could be shown in a figure.

Agreed, we have provided a new figure (Figure 1) that illustrates the change in linearity of the Ct values between the two PCR assays as the A/E times are shortened.

  1. Figure 1. The figure is not explained sufficiently. What is shown in the two curves? What is shown on the ordinate?

We have provided two new Figures to illustrate our data with clear X- Y-axis explanations.

Reviewer 3 Report (New Reviewer)

Comments and Suggestions for Authors

In this paper, the authors investigate the reaction conditions of an ultrafast plasma polymerase chain reaction (PCR) using the Kimera P-IV, VCSEL 4-channel plasmonic PCR instrument (Nexless Healthcare, 102 Montreal, QC, Canada). Their aim is to compare the performance of two PCR mixes, namely Luna Universal qPCR & RT-qPCR and Solis FAST SolisGreen qPCR Mix (Solis Bio-65 dyne, Tartu, Estonia), under identical conditions. The obtained results present promising outcomes.

However, there are a few observations to consider:

  1. The title seems somewhat exaggerated and may not accurately represent the experimental scope of this work.

    • The first part, "Diagnostic Speed and Accuracy Redefined...", may overstate the findings since the evaluation was limited to a single synthetic gene. To truly redefine diagnostic speed and accuracy, a broader range of genes, possibly from clinical samples, should be assessed, considering the potential variability in performance.
    • The second part, "...and the Evolution of Taq-Polymerase Engineering," merely describes a selection of master mixes chosen for convenience.
  2. The phrase "PCR assay for virtually any nucleic acid target" (Line 30) appears presumptive and lacks scientific rigor.

  3. Clarification regarding the orientation of the mentioned oligonucleotides (Lines 83 and 84) is necessary for better understanding.

  4. While the tables and figures effectively present the obtained results, it's noted that while the study is interesting for the scientific community, several other works have already explored the potential of plasmonic PCR as an ultrafast diagnostic alternative. Therefore, it's essential to emphasize the distinctive aspect of this study.

  5. To transition from a theoretical and controlled setting to real-world applications, validation using clinical samples is indispensable.

In summary, while the study presents promising findings regarding the evaluation of reaction conditions for ultrafast plasma PCR, attention to the aforementioned points would enhance the clarity and impact of the paper within the scientific community. 

Author Response

Reviewer 3

In this paper, the authors investigate the reaction conditions of an ultrafast plasma polymerase chain reaction (PCR) using the Kimera P-IV, VCSEL 4-channel plasmonic PCR instrument (Nexless Healthcare, 102 Montreal, QC, Canada). Their aim is to compare the performance of two PCR mixes, namely Luna Universal qPCR & RT-qPCR and Solis FAST SolisGreen qPCR Mix (Solis Bio-65 dyne, Tartu, Estonia), under identical conditions. The obtained results present promising outcomes.

However, there are a few observations to consider:

  1. The title seems somewhat exaggerated and may not accurately represent the experimental scope of this work.

Thank you, we agree that the title should be changed.

The first part, "Diagnostic Speed and Accuracy Redefined...", may overstate the findings since the evaluation was limited to a single synthetic gene. To truly redefine diagnostic speed and accuracy, a broader range of genes, possibly from clinical samples, should be assessed, considering the potential variability in performance. The second part, "...and the Evolution of Taq-Polymerase Engineering," merely describes a selection of master mixes chosen for convenience.

We are suggesting a more appropriate title “Assessing Different PCR Master Mixes for Ultrarapid DNA Amplification: Important Analytical Parameters.”

The phrase "PCR assay for virtually any nucleic acid target" (Line 30) appears presumptive and lacks scientific rigor.

Thank you for this comment, agreed, that we removed this claim and specified to target we used.

Clarification regarding the orientation of the mentioned oligonucleotides (Lines 83 and 84) is necessary for better understanding.

The primer-sequence character presentation is 5’-3’ direction. We will add 5’ and 3’ symbols.

While the tables and figures effectively present the obtained results, it's noted that while the study is interesting for the scientific community, several other works have already explored the potential of plasmonic PCR as an ultrafast diagnostic alternative. Therefore, it's essential to emphasize the distinctive aspect of this study.

Here is short summary of experimental data presented in our MS, which are emphasizing distinctive aspect of our study:

(Claim 1) Our analysis showed that post-PCR Tm analysis could effectively distinguish shorter amplicons (“primer-dimers”) from the intended ones. The Tm values for intended amplicons were consistently few degrees higher than for non-intended products, a distinction that could be incorporated into clinical data analysis algorithms:

Using 10-times serial dilutions of Positive Control DNA (Covid-19 N gene, IDT DNA), we evaluated the Luna and Solis PCR assays across a 4 log10 range of input DNA. This analysis involved measuring cycle threshold values (Ct) and temperature melting profiles (Tm) of post-PCR products. We varied annealing/elongation times (AE) from 15, 5, to 1 second to identify performance under extremely short cycling conditions. Replicates were done in triplicate for each target concentration, and in cases of lower input DNA, two distinct Tm peaks were occasionally observed, separated by a backslash in Appendix A, Table 1, under Tm columns. At 15 seconds AE time, similar Ct and Tm results were found between Luna and Solis assays, particularly at higher input concentrations. However, differences became apparent and consistent when AE times were less than 15 seconds or initial DNA target concentrations were extremely low. For the longest AE time (15 seconds), a linear change of Ct values was expected and observed with the logarithm of input target concentrations (Coefficient of correlation R=0.98). This linearity declined with shorter AE times, affecting the quantitative analysis, as non-fully elongated DNA fragments contributed to the Ct value along with intended amplicons. However, the post-PCR Tm parameters could effectively distinguish shorter amplicons from the intended ones. The Tm values for intended amplicons were consistently few degrees higher than for non-intended products, a distinction that could be incorporated into clinical data analysis algorithms.

(Claim 2): There is a difference in Taq Polymerase Performance among different PCR master mix vendors:  A notable difference was observed in the capacity of the two Taq polymerases to produce amplicons at 1 second AE time. The Luna assay consistently failed to generate intended PCR amplicons under these conditions, whereas the Solis assay succeeded. This suggests that mutations in the SolisFAST Solis Green Taq polymerase may enhance sensitivity, possibly due to increased PCR reaction speed.

(Claim 3): The plasmonic PCR reaction shows excellent Reproducibility and Accuracy: To assess PCR reproducibility, we calculated the Coefficient of Variation (CV) of Tm values across triplicate experiments. A lower CV indicates greater reproducibility. For example, the CV for Tm values in the Solis assay was only 0.6% across all AE times and DNA template concentrations. Accuracy, defined as the ability to discriminate between Tm profiles of intended amplicons, primer-dimers, and negative controls, was evaluated by com-paring the number of correct "predictions" to the total number of "predictions". Our findings show that the Tm parameter is crucial for estimating a "positive PCR" and guarantees maximum analytical accuracy within the limits of detection.

(Claim 4): There are important Implications for future designs of TaqMan Probe-based Assays: The Solis assay's engineered Taq polymerase retains 5’-3’ exonuclease activity, allowing the use of TaqMan hydrolysis probes in PCR multiplexing. This could significantly facilitate assay development compared to the intercalating dye-based assays used in this study. It would be beneficial to investigate whether the new Solis FAST® SolisGreen® qPCR enzyme exhibits faster elongation (processivity) speed or an optimized change in affinity toward the template (Kon/Koff), contributing to ultrafast PCR capabilities.

To transition from a theoretical and controlled setting to real-world applications, validation using clinical samples is indispensable. In summary, while the study presents promising findings regarding the evaluation of reaction conditions for ultrafast plasma PCR, attention to the aforementioned points would enhance the clarity and impact of the paper within the scientific community.

We thank you Review 3 for the relevant comments and will add the final/last sentence: “For future  real-world applications,  the validation using clinical samples is indispensable”

Round 2

Reviewer 2 Report (Previous Reviewer 1)

Comments and Suggestions for Authors

For now, the presented data in the manuscript is still too weak to support the publication of this paper. Moreover, the added two figures are low quality. Noteworthy, the simply comparason of two commercialized PCR kits and optimize the PCR conditions are not innovative, which are just labor-work and can be accomplished by any normal undergratudate. The authors should show your intellectual contributions.

Comments on the Quality of English Language

none

Reviewer 3 Report (New Reviewer)

Comments and Suggestions for Authors

In this form, the paper can be a valuable tool for the scientific community and represents a significant contribution to the development of precise, rapid, and effective diagnostics. However, it is necessary to perform validation in a relevant environment with clinical samples

This manuscript is a resubmission of an earlier submission. The following is a list of the peer review reports and author responses from that submission.

Round 1

Reviewer 1 Report

Comments and Suggestions for Authors

The authors compared Luna Universal and Solis FAST®SolidGreen, two Taq-DNA ultrafast enzymes with 5'-3' exonuclease activity, for plasma® ultrafast PCR reactions, to regulate elongation time and N gene concentration parameters, and to evaluate PCR amplification results by Ct values and melting curves, redefining the speed and accuracy of PCR diagnosis. The results showed that the Ct and Tm results were similar at 15 seconds of annealing, and that the Luna assay was unsuccessful due to the extremely short annealing and elongation times of one second, while the Solis assay still demonstrated successful PCR. The use of Tm will guarantee maximum analytical accuracy at a value of 1 that reliably differentiates positive and negative PCR test results within the detection limit. The techniques used have been previously reported, but the correlation of Tm with PCR amplification results is rarely mentioned. Therefore, I support publishing it in diagnostics. Here are some of the key comments from this work:

1. The authors must do a comprehensive review for the ultrafast plasmonic PCR in the Introduction, since the ultrafast PCR has been proposed and developed for many years. Some critical studies reported about the ultrafast PCR should be referred to, such as Light: Science & Applications 2015, 4, e280; Trends in Biochemical Sciences 2020, 45 (2), 174-175.

2. In this work, the photothermal material (gold nanorod), ultrafast enzyme and PCR instrument are all commercially available, and the experimental design is not innovative.

3. The major concern is that the presented data is too little to support the results. The authors thus are suggested to add more data, including nanomaterials characterization, experimental condition optimization, ultrafast PCR performance etc.

4. The supplementary materials were linked to a company website and a Thesis of Master Degree, which is not proper for the published papers.

Comments on the Quality of English Language

None

Reviewer 2 Report

Comments and Suggestions for Authors

 diagnostics-2804355

Diagnostic Speed and Accuracy Redefined: Ultra-Fast Plasmonic PCR and the Evolution of Taq-Polymerase Engineering.

Real-time plasmonic PCR of the Covid-19 DNA N gene positive control DNA with 10-10,000 copies of input and 1-15 sec. of annealing/elongation (in triplicate) was performed using two different master mixes.

The following points should be considered:

1. M&M, line 75-78. Why is this relevant for M&M? It may be relevant for the introduction, but not in this section.

2. M&M, line 92 and elsewhere. Reference 16 is a master thesis, which is hard to access for future readers. If there are relevant information in this thesis, it should be included in the paper and the author of the thesis included as co-author.

3. Results and discussion, line 109-111. This is not described in M&M.

4. Results and discussion, line 113-114. There are no results documenting this.

5. Results and discussion, line 117-120. Please refer to table 1.

6. Results and discussion, line 141-142. The linear change could be shown in a figure.

7. Figure 1. The figure is not explained sufficiently. What is shown in the two curves? What is shown on the ordinate?
